# Moving analytical ultracentrifugation software to a good manufacturing practices (GMP) environment

**Alexey Savelyev[1], Gary E. Gorbet[2], Amy Henrickson[3], Borries Demeler[1,2,3]** *

**1** University of Montana, Dept. of Chemistry, Missoula, Montana, United States of America, **2** AUC Solutions, Houston, Texas, United States of America, **3** University of Lethbridge, Dept. of Chemistry and Biochemistry, Lethbridge, Alberta, Canada

* demeler@gmail.com

## Abstract

Recent advances in instrumentation have moved analytical ultracentrifugation (AUC) closer to a possible validation in a Good Manufacturing Practices (GMP) environment. In order for AUC to be validated for a GMP environment, stringent requirements need to be satisfied; analysis procedures must be evaluated for consistency and reproducibility, and GMP capable data acquisition software needs to be developed and validated. These requirements extend to multiple regulatory aspects, covering documentation of instrument hardware functionality, data handling and software for data acquisition and data analysis, process control, audit trails and automation. Here we review the requirements for GMP validation of data acquisition software and illustrate software solutions based on UltraScan that address these requirements as far as they relate to the operation and data handling in conjunction with the latest analytical ultracentrifuge, the Optima AUC by Beckman Coulter. The software targets the needs of regulatory agencies, where AUC plays a critical role in the solution-based characterization of biopolymers and macromolecular assemblies. Biopharmaceutical and regulatory agencies rely heavily on this technique for characterizations of pharmaceutical formulations, biosimilars, injectables, nanoparticles, and other soluble therapeutics. Because of its resolving power, AUC is a favorite application, despite the current lack of GMP validation. We believe that recent advances in standards, hardware, and software presented in this work manage to bridge this gap and allow AUC to be routinely used in a GMP environment. AUC has great potential to provide more detailed information, at higher resolution, and with greater confidence than other analytical techniques, and our software satisfies an urgent need for AUC operation in the GMP environment. The software, including documentation, are publicly available for free download from Github. The multi-platform software is licensed by the LGPL v.3 open source license and supports Windows, Mac and Linux platforms. Installation instructions and a mailing list are available from ultrascan.aucsolutions.com.

**Data Availability Statement:** All relevant data are within the manuscript and its Supporting Information files.

**Funding:** Funding for this work was provided by: National Institutes of Health grant 1R01GM120600, NSERC DG-RGPIN-2019-05637, CIHR foundation grant (FDN 148469), and Canada 150 Research Chairs program C150-2017-00015 (All to BD). The funders had no role in study design, data collection and analysis, decision to publish, or preparation of the manuscript.

**Competing interests:** I have read the journal's policy and the authors of this manuscript have the following competing interests: Gary Gorbet and Borries Demeler declare ownership interests in AUC Solutions, Houston, Texas. AUC Solutions provides consulting services and UltraScan software support for analytical ultracentrifugation users.

This is a *PLOS Computational Biology* Software paper.

## Introduction

Analytical ultracentrifugation (AUC) is a nearly one hundred year old technique for the solution characterization of colloidal molecules. Invented by Nobel laureate Theodor Svedberg, its original application was in protein science. Today, AUC is used to study a broad range of other molecules, including synthetic polymers [1], nucleic acids [2], nanoparticles [3, 4], carbohydrates, lipids and detergents [5], and macromolecular assemblies and complexes. AUC is no longer only used in fields like biochemistry and structural biology, but has gained a firm place in material science [6], physics, and most notably biopharma. Here, AUC aids in the development of formulations by monitoring forced degradation studies to quantify the amount of aggregates, with quality control, research and development of novel therapeutics, and characterization of interactions between biopolymers in pharmacological applications [7]. Good Manufacturing Practices (GMP) describes a set of operating and instrument standards, process control requirements and documentation that, when followed, provide assurances to the consumer and regulatory agencies that best practices are followed in the laboratory environment, and that results are validated and can be audited, quality standards are observed, and the risks in pharmaceutical production are minimized [7]. GMP laboratory environments are an important component of pharmaceutical production and are essential for the quality control processes in the characterization of pharmaceutical products [8]. High demands are placed on these processes in terms of resolution, reproducibility, dynamic range, accuracy, precision, and compliance. GMP attempts to assure that products do not vary between different lots and are of a consistently high quality, all of which require that the quality control processes are stringent and follow a minimum standard. When it comes to pharmaceutical products, the consumer's safety is a primary concern, and GMP provides the rules a laboratory needs to follow to meet regulatory approval.

Among other solution characterization techniques for molecules such as size exclusion chromatography, dynamic and static light scattering, and field flow fractionation, AUC offers a particularly attractive characterization potential for biopharma applications. There are several compelling reasons: 1. Due to the hydrodynamic separation of all molecules based on their size, densities, and anisotropies, AUC provides unmatched resolution in these parameters. 2. As a first-principle technique, AUC does not require any standards. All flow occurring in the ultracentrifugation cell is exactly described by a partial differential equation, and recent advances in computing and optimization have made great strides in further enhancing the information content and throughput achieved in the analysis of AUC data. 3. AUC has a remarkable dynamic size range which can be exploited through variable rotor speed and extended through variable buffer density and viscosity, and it has high sensitivity, reaching from the picomolar to high micromolar concentration range. 4. Multiple optical systems (UV-visible absorbance, multi-wavelength UV-visible absorbance, fluorescence, and Rayleigh interference optics) offer a broad range of orthogonal measurement possibilities and further expand the applicability of this technique. In our present work we discuss the challenges and solutions to address the demands of AUC for a GMP environment.

## Challenges for GMP validation of AUC

GMP validation of AUC faces hurdles on several levels that need to be addressed before AUC is ready to be moved into a true GMP environment:

1. Sample handling and AUC cell loading is a manual process, requiring dexterity to achieve reproducible loading volumes and proper alignment of AUC cells in the rotor. Both are subject to human error that is difficult to control even under the best of circumstances.

2. The AUC instrument is a highly sophisticated instrument, and its operation is subject to a large number of variables that, when not properly controlled, can alter outcomes in unanticipated or undetectable ways. The instrument has a very large number of components that all should be validated independently, a difficult proposition given that each may come from a different supplier, or suppliers could change and components and their inter-operability would require re-validation.

3. Traditional data acquisition software is disconnected from the data analysis process and requires manual transfer of data between multiple computers, with additional manipulation of the collected data before analysis can occur, introducing a weakness in the chain of data custody, impairing the audit trail, and potentially allowing falsification of the primary research data.

4. Currently, all data analysis is performed with software that has not been documented and validated for GMP use, much of it is also closed-source freeware [9, 10, 11], where any audit of the source code is impossible and repeat analysis of the same data may result in different answers because there is no guarantee that the same analysis parameters are reused and no assurances are given that the software follows stringent GMP documentation requirements. This is particularly troublesome for quality control applications and the study of biosimilars where strict repeatability is required.

5. Important parameters that could affect the analysis results are not routinely checked for consistency. These parameters include the number of scans in an experiment, the duration of the experiment, loading volume and column length, all of which affect resolution and information content, as well as loading concentration which can affect mass action of reversible systems and non-ideality parameters, as well as recording wavelengths which can lead to incorrect radial position reporting due to chromatic aberration.

6. The data quality is not checked for consistency, and data editing, analysis algorithms, and analysis parameters are not checked for appropriateness and left up to the user to change at will. Analysis parameters influencing data interpretation such as grid sizes, grid resolution, edited data ranges, buffer corrections, as well as hardware parameters such as rotor stretch corrections influence analysis results, and are not controlled for consistency.

7. Reporting is not an automated process; it requires manual processing of data, and it is based on a subjective interpretation of results obtained from a black-box analysis system and does not include standard metrics for comparison that can be automated and applied without user or operator bias.

Despite these challenges, most, if not all, relevant variables in the data management can be sufficiently controlled in the analysis software. Consistent and reproducible routine AUC analysis can be achieved by implementing appropriate workflow validation, instrument diagnostics, and greater automation in the data acquisition and analysis processes. This ability is greatly aided by the arrival of the new Optima AUC analytical ultracentrifuge from Beckman Coulter (Optima), which features important upgrades that provide not only much higher data quality and density, but also contemporary technology and design choices that greatly facilitate the operation of the analytical ultracentrifuge in an automated GMP environment. Most importantly, the instrument can be networked over Ethernet, and data acquisition is

accomplished by a computer configured with the open source Linux operating system, utilizing a PostgreSQL relational database for storage of experimental data, run profiles, and system status information. Unlike the older Proteomelab instruments, which stored data in ASCII format, experimental data are now stored in 8 byte double precision format in the database, which assures that the full accuracy of the detector is preserved. Binary data also offers much faster input/output and network transfer speeds compared to the ASCII format used on the Proteomelab. The ability to directly interact with the PostgreSQL database enables additional advantageous features that are important for GMP operation: 1. remote submission of experimental analysis profiles; 2. automatic operation of the instrument; 3. remote monitoring; 4. fully automated data transfer to the analysis computer in the native format; 5. access to fine-grained, real-time system data (time, rotor force integral, temperature) with 1 second resolution, and; 6. implementation of audit trails for data access at any level.

A significant challenge for analytical ultracentrifugation performed in a GMP environment is that the operation of the analytical ultracentrifuge involves multiple hardware components that are responsible for experimental observables and parameters that directly affect analysis results. Included in this list are rotor speed, recorded time and centrifugal force integrals, temperature, radial calibration, light intensity, optical alignment and focus, chromatic aberration, and stretching or contraction of the rotor in response to rotor speed changes. When operating under GMP conditions, it would be desirable if these components could be independently validated; however, this would be cost-prohibitive and impractical. For example, when the manufacturers of individual hardware components are exchanged, or operating software components are updated, each instrument would have to be independently revalidated. To address this challenge, we propose a strategy by which the instrument's proper operation is validated by performing a series of diagnostics that can either be readily executed by the operator or integrated into an automated analysis and reporting structure to assure the integrity of basic functionality. Such tests include verification of radial calibration, light intensity, chromatic aberration, and rotor stretch. A validated, speed-appropriate standard can be included in each GMP experiment to check the validity of the data acquisition process, and to validate experimental observables (rotor speed, temperature, recorded time and centrifugal force integrals). The standard itself will be validated with orthogonal GMP methods. A reference experiment for the standard will be compared upon completion for essential features such as sedimentation- and diffusion coefficient distributions to validate composition, root mean square deviation (RMSD) of the fit, average light intensity throughout the experiment, loading concentration and loading volume. If the comparisons with the reference data fall within the acceptable error tolerances defined in the reference experiment, the instrument can be considered validated for all variable parameters of the instrument, and since AUC is a first-principle technique, this ensures that the analysis can solely be based on the instrument observables.

Several instrument properties influencing experimental observables can be readily validated by the user without any instrument modifications. We have designed new calibration tools and developed novel data acquisition and analysis software that together will provide GMP support for analytical ultracentrifugation. These include a new calibration disk that can be used in place of the counterbalance in the 4th or 8th rotor hole position to enable the delay calibration for absorbance or intensity experiments at any speed to determine the rotor angle, and to validate the radial calibration setting. It also has features designed to measure the rotor stretch function of any rotor, as a function of speed, and to measure radial position errors resulting from chromatic aberration as a function of wavelength. This work was recently described in [12], and has been implemented in the latest version of UltraScan [13, 14], which associates a unique stretch profile with each rotor entered into the LIMS database [15] which

provides an automatic stretch corrections during data analysis, and appropriate chromatic aberration corrections during data import for Optima AUC instruments.

## Design and Implementation

### A. Time and $\omega^2 t$ Integrals

Unlike the older Beckman Proteomelab XLI/XLA data acquisition software, which only recorded scan times and $\omega^2 t$ integrals for each collected scan, UltraScan provides a scan-independent timestate object, which is part of the openAUC data format [16] and is further discussed in [17]. In the Optima, it provides a second-by-second accounting of set and actual rotor speed, speed step, time, $\omega^2 t$ integrals, temperature and scan frequency, starting with the first second of acceleration. Using the 1-second counter of the PostgreSQL database, the time of recorded scans, their $\omega^2 t$ integrals, and their rotor speed can be verified by a separate timer for accuracy. On two Optima instruments at the University of Lethbridge in Alberta, one instrument at Mayo Clinic in Rochester, Minnesota, and one instrument at Hormel Institute at Austin, Minnesota, we have not found any deviations from the recorded scan times and have found the instruments to be accurate within the resolution of detection. Nevertheless, the UltraScan timestate object can be examined in a graphical user interface to assure accuracy (see S1 Fig).

### B. Validated standards

As mentioned above, it is impractical to validate each hardware component of the entire instrument for GMP operation. Since AUC is considered a first-principle technique, it is possible to validate the AUC's output with a known standard. If the AUC experiment conducted with the standard(s) results in the expected values, the instrument can be considered validated. As it is accepted procedure to periodically test the validation of other GMP methods like high performance liquid chromatography (HPLC) by measuring an appropriate standard, the same procedure can be used for AUC as well. There are several requirements for an acceptable AUC standard: 1. The standard needs to be sufficiently stable and provide sufficient shelf life during which all measurable properties do not change. 2. Characterization of the standard by orthogonal GMP methods must be possible. 3. The standard must exhibit properties that address all the parameters measurable and detectable by AUC (sedimentation coefficient, diffusion coefficient, partial specific volume, anisotropy, molar mass, hydrodynamic radius, relative concentration, and concentration dependent non-ideality). 4. It must be possible to characterize these parameters over the entire dynamic range of the instrument. 5. The standard must be available with sufficient purity, and it must be possible to manufacture this standard with great reproducibility.

We propose that the ideal standard for GMP-AUC is double-stranded DNA (dsDNA). It satisfies all of the above requirements: 1. dsDNA is one of the most stable biopolymers available, and remains unchanged for years under appropriate storage conditions. 2. dsDNA molecules can be characterized with other GMP methods such as sequencing, size exclusion chromatography, dynamic and static multi-angle light scattering, fluorescence correlation spectroscopy, and gel electrophoresis. 3. By changing the length of the dsDNA sequence, discrete sized molecules can be created of arbitrary size, which modulates their molar mass and their diffusion coefficient, hydrodynamic radius, and frictional ratio. dsDNA exhibits a strong anisotropy signal as a function of length and molar mass, and its partial specific volume, anisotropy, and non-ideality properties can be easily modulated by using buffers with different ionic strengths [18]. 4. A set of dsDNA standards will have different lengths, stored under select buffer conditions, which can be utilized alone or in combination at different ratios to

generate standards for different rotor speeds. Staining DNA with fluorophores like SybrGreen imparts absorbance in the visible range to extend the utility of the standard for wavelengths outside of the UV range. 5. dsDNA can be readily produced using recombinant cloning technologies to assure reproducible sequences, and vectors containing different lengths of dsDNA fragments can be readily mass produced using bacterial cultures. Size exclusion chromatography and gel electrophoresis can be used to obtain ultra-pure preparations of the dsDNA standards at different lengths. dsDNA molecules of varying lengths can be mixed at varying ratios to produce standards that validate limits of detection (LOD) and limits of quantification (LOQ). Multiple instruments can be used to establish precision limits for AUC measurements and calibrate the utility of dsDNA polymers for the use in AUC standards.

## C. Data acquisition software

In a significant departure from previous instrument designs, the Optima has been equipped with a novel data acquisition system that lends itself to supporting important aspects of GMP operation. Unlike its predecessor, the ProteomeLab XLA/XLI instruments, the software responsible for data acquisition uses reliable and high performance open source tools, such as the Linux operating system, coupled with an open source relational database (PostgreSQL), and the Apache Tomcat web server. This approach follows in the footsteps of the openAUC architecture proposed earlier [16], and permits direct interfacing with the instrument over ethernet using alternative third-party software solutions like UltraScan. This avenue offers maximum flexibility and fosters development of new and interesting tools that can explore novel features like multi-wavelength experiments [19, 20, 21, 22, 23, 24] and GMP capable data acquisition software, discussed here. Using the network-capable Linux tools provided with the Optima, our laboratory has extended the UltraScan-III AUC analysis software [13, 14] and the UltraScan Laboratory Information Management Systems (UltraScan-LIMS) framework [15] to interface directly with the Optima for experimental design, data acquisition, and remote experiment monitoring. The goal for this software is to greatly simplify operation of the analytical ultracentrifuge and to automate all steps in the data acquisition and data analysis process without sacrificing flexibility. The newly developed software encompasses a number of core components, each of which is discussed further below. These components are unified into a single and intuitive graphical user interface which largely automates the entire workflow, starting with experimental design, submitting experimental protocols to the Optima, monitoring data acquisition, retrieving experimental data, editing the data, analyzing the data according to standardized profiles, comparing the results to a reference dataset, and preparing a human readable report for evaluating the results. The entire workflow is controlled by an 'autoflow' supervisor, which keeps track of each stage of the data collection and analysis process for one or more Optima instruments. The autoflow process is followed by one or more remote computers that can re-attach to the process at any stage to monitor progress with data acquisition or to initiate a new experiment when an instrument is not in use.

 **C1. Database layout.** An important aspect of GMP operation is a verifiable data management policy where data ownership is tightly controlled, data are stored in well-defined structured linkages where all data transactions can be audited and timestamped, a proper chain of data custody can be assured, and access by authorized individuals can be verified. Such GMP operation is greatly alleviated by the use of a relational database. Such databases enforce linkages between data tokens, and can specify allowed operations by using 'stored procedures'. The UltraScan-III software uses its own relational database, MariaDB [25], an open source database which is derived from the MySQL database engine. In UltraScan, all relevant details from the experiment are stored in a relational database, using stored procedures. Stored

procedures allow tight control of how data are used, deleted, modified, and accessed. Data access is tightly controlled by multiple user levels to fine-tune access permissions and safeguard data. Importantly, the database can enforce 'foreign key constraints', a mechanism that can be used to prevent inadvertent deletion of associated data, or used to enforce a cascading deletion operation, assuring removal of otherwise orphaned nodes in the database. Database functionality within UltraScan is provided through the desktop client software [13], as well as through the web-based UltraScan Laboratory Information Management System (UltraScan-LIMS) [26]. The new features described here address the use of UltraScan for GMP applications and is collectively referred to as UltraScan GMP.

The basic concept of the UltraScan-LIMS database is to start with a detailed experimental design, termed a 'protocol', which contains all relevant details about the experiment to be performed. This information is entered prior to the experiment at a time when all experimental details such as the loading pattern of cells, the rotor serial number, the types of windows and centerpieces used, etc. are readily available. Using hierarchical relationships in the database, each 'protocol' is connected to one or more 'experiment', which in turn is linked to a data owner (the 'investigator'), a technician (the 'operator') who performs the experiment, a 'rotor', and a set of 'cells'. Each experiment is performed at a specific temperature, and has a user-defined delay for temperature equilibration. An experiment is identified by its 'experimentID' and a verbose 'label'. Each cell contains one or more 'channels' which can contain separate samples, identified by 'solutions'. Solutions contain 'buffers' composed of one or more buffer components and one or more 'analytes' at specified molar ratios. 'Solutions', 'buffers' and 'analytes' have defined spectra recorded in the database in the form of intrinsic molar extinction coefficients, as well as automatically (in the case of protein sequences) or manually defined partial specific volume, density, viscosity, and pH values. Experiments are also linked to 'projects' which represent digital laboratory notebooks that track the history and ancillary details of an experimental process. Experiments are performed in 'rotors' which inherit one or more stretch 'calibrations' that contain the stretch profiles of the rotor at different loading weights and/or temperatures. Experimental data are also associated with an instrument and a laboratory where the experiment was performed. Experiments have one or more speed step associated with them (recorded in the timestate object, derived from data stored in the database), which specify the rotor speed of the step, as well as the scanning frequency, the stage delay and scan interval for each optical system. Experimental data are contained in a 'triple', which defines the cell, channel, and wavelength for each dataset. In the case of a multi-wavelength experiment, a channel can contain multiple triples, one for each wavelength. Cells are identified by their centerpiece type, sector geometry, pathlength and material, as well as the window type. Each channel can be scanned with either UV/visible absorbance optics or Rayleigh interference optics, and wavelength ranges can be specified for each cell.

Once experimental AUC data are collected, analysis starts with an 'edit profile', which defines the initial meniscus position, the bottom of the cell position, which is derived from the centerpiece geometry, the rotor stretch calibration, the right data limit, and the scans to be used for subsequent data fitting. Each triple can inherit one or more edit profiles. Fitted data result in 'models', which are linked to the edit profile, the analysis method, and the triple associated with the fitted data. These models represent spreadsheets that contain the uncorrected and 20,W corrected sedimentation and diffusion coefficients, frictional ratios, molar masses, and partial concentrations for each species fitted in the analysis. For reacting systems, models also contain equilibrium constants for each reaction, as well as kinetic rate constants, where applicable. All models can be passed between analysis methods and visualization and reporting modules to generate graphs and detailed reports. Reports are also linked to the associated triples, or a single experiment ID when global or combined analyses are performed that do not

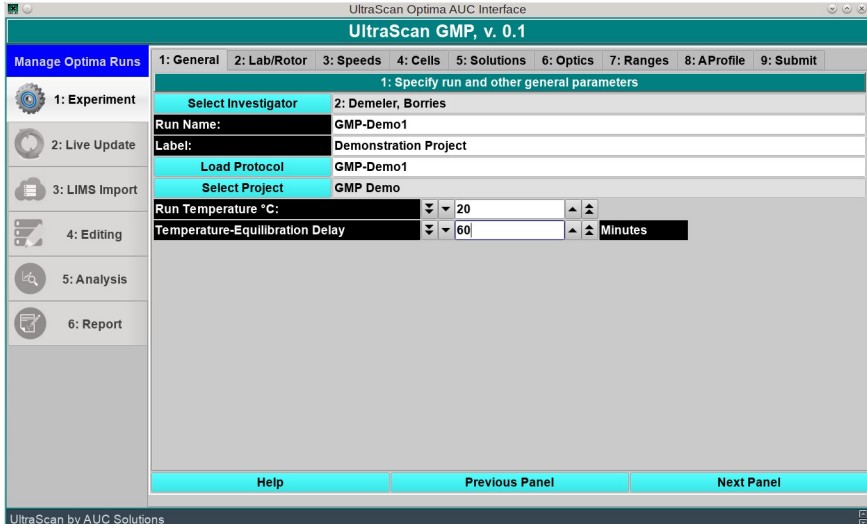

**Fig 1. UltraScan GMP opening screen for the experimental analysis profile definition.**

have a unique triple as a parent [13]. Together, these data tokens uniquely define an AUC experiment's parameters, the analysis results, and reports.

**C2. UltraScan GMP graphical user interface.** The UltraScan GMP software provides a multi-platform graphical user interface for user interaction that is available for Windows, Macintosh, and Linux computers. The goal for this interface is to allow a technician with minimal training in biophysics to successfully operate the instrument in a GMP environment and monitor data acquisition as well as the analysis workflow. A separate interface for a senior scientist provides easy access to all functions needed for the development of an experimental design and analysis workflow. The type of interface presented is dependent on the user level of the individual logged into this system. Permissions are set based on the user level when the user logs in during the initial opening of the program. The user level then controls permissions for all subsequent actions, including preventing the modification of read-only elements in the database.

The program is organized into 6 distinct steps, shown in a vertical tab row on the left (see S2 Fig). The program will automatically cycle from one tab to the next and prompt the user only when any interaction is needed. At the top, a horizontal tab may offer sub-tasks, depending on each step that needs to be completed sequentially. If the program is opened during an ongoing experiment, the program will detect the status of the experiment running on any of one or more instruments connected to this system, and present the option to either re-attach to a currently running experiment, or to design and submit an experiment to a machine that is currently unoccupied.

**C3. Experimental analysis profile definition.** In the first step, the experimental analysis profile is either loaded by a technician in read-only mode, allowing only the experiment name to be changed, or a new design is created (Fig 1) by a user with elevated privileges. This way, GMP procedures are exactly replicated in a repeat experiment, without any possibility of critical parameters being accidentally changed. The only field that can be changed for a stored profile is its run name, which needs to be adapted for a new lot or different date of the experiment. The UltraScan software prevents the user from using the same name twice for different experiments. In this screen, the run temperature and the delay for rotor temperature equilibration are also set. In the next screen, the laboratory in which this experiment is carried

out is identified, and the instrument, rotor and rotor stretch calibration, operator, and the type of experiment are selected. The program will check the current connection status of the instrument (see S3 Fig).

Single- or multi-speed experiments are defined on the next screen. For each speed step, initial delay, scanning time, and scan interval can be set. The total number of scans are automatically calculated based on the length of active scanning time for a single scan, which is speed dependent. The user can readily adapt the duration of a particular speed step to achieve a desired number of scans with user-selected scan delays. A minimum initial scan delay up to the first scan is a function of rotor acceleration time as well as the internal calibration steps for the science modules of the Optima, which can be increased by the user. Minimum initial scan delays, as well as scan times, differ for the Rayleigh interference and the UV/visible absorbance optical systems, hence separate controls are available for each system, in case both optical systems are used in parallel for an experiment. A screenshot of the speed setting screen is shown in S4 Fig.

In the next screen, cells to be scanned are identified by the centerpiece and windows used. If one cell in the rotor is configured with quartz or sapphire windows, the same window and centerpiece configuration is automatically chosen for the opposite cell to assure weight balance. UltraScan is able to utilize all rotor positions for data acquisition in UV/visible detection mode, and a counterbalance does not need to be located in hole 4 (An60Ti rotor) or 8 (An50Ti rotor), as long as a 3,000 rpm radial calibration has already been performed with the counterbalance in place in a previous calibration run (see S5 Fig). For GMP experiments, the calibration centerpiece [12] can be used instead of a counterbalance to validate radial calibration, and both a 2-channel centerpiece or the calibration centerpiece can be used for the delay calibration.

In the next screen, the solution for each channel needs to be chosen (S6 Fig). These are selected from pre-defined solutions in the database that contain a buffer solution and one or more analytes (proteins, nucleic acids, carbohydrates, or others). An analyte in the database provides a partial specific volume, extinction coefficients and absorbance spectra to a solution, the buffer components of the solution specify the density and viscosity of the solution. These values are automatically used for hydrodynamic corrections to the analysis results.

In the "Optics" screen (S7 Fig), individual cells are associated with the available optical systems in the instrument. Use of a sample cell in holes 4 (for An60Ti rotors) or 8 (for An50Ti rotors) prevents usage of the interference optical system, since Rayleigh interference measurements require a counterbalance to perform the delay calibration. By default, the operation of the Optima acquires intensity data in UV/visible detection mode. Avoiding absorbance detection is preferred, and intensity mode is enforced in UltraScan to double the capacity of the instrument, by allowing for samples to be run in both channels of the cell, and to decrease stochastic noise contributions [27]. The definition of radial measurement ranges and wavelength ranges in UV/visible detection mode is performed in the next screen (S8 Fig). To facilitate wavelength selections, UltraScan GMP offers a simple scripting language, as well as a wizard to select desired wavelengths or wavelength ranges (S9 Fig).

An important innovation in UltraScan GMP is the ability to define analysis profiles. These profiles become part of the experimental design stored in the database, and must be defined before the experiment is performed. For GMP operation, it is critical that all aspects of a GMP analysis are reproducible and that all conditions of a replicate experiment or comparison experiment are identical to the extent that this can be achieved. The variables that can be controlled here do not only extend to the rotor speed, temperature, and experimental duration, but include many other conditions, such as loading concentration, loading volume, solution properties, data owner, optical systems, window type, scan frequency, scanning wavelength,

centerpiece geometry, rotor stretch properties, as well as the instrument on which the data collection is performed. A second requirement is that the analysis parameters with which the experiments are analyzed are replicated identically for each sample. For example, comparisons between a market brand drug and a generic biosimilar formulation will require identical analysis workflows of the brand name drug and the biosimilar formulation in order for the comparison to be valid. Our approach is to not only define specific analysis parameters that need to be used during fitting of experimental data, but also to define analysis tolerances that reflect intrinsic confidence intervals of the technique, given the available instrumentation.

These tolerances need to be defined for each experimental system, since they are dependent on rotor speed, optical properties of the sample, optical systems and settings, and other factors that are instrument related. As is shown in S10 Fig, each channel is assigned a concentration loading ratio. By default, this ratio is unity, indicating the same concentration as was used in a reference experiment is applied here. Maintenance of identical concentration in biosimilar comparison experiments is important, since some samples change association state as a function of mass action, which changes sample composition. If mass action is to be monitored, multiple concentration ratios should be examined. Due to experimental error and micro-variations in the centerpiece pathlength, the exact same concentration may not be reproducible, therefore a tolerance value within which the experiment is considered to be repeatable is assigned. This value will guide the reporting module, and deviations from this tolerance will be flagged in the report. The loading volume of a sample in a velocity experiment is proportional to the signal-to-noise ratio and the resolution that can be obtained. The longer the solution column, the higher the resolution with which different sedimenting species can be resolved. Therefore, it is important that the loading volume is maintained across comparison experiments. To account for pipetting error, a loading volume tolerance value can be specified. Like the solution column length, the data range to be evaluated is also proportional to the resolution observed, hence the right data limit must be defined as well. Rotor-stretch dependent boundary conditions for solutions of the Lamm equation are calculated from the rotor stretch calibration profile, and independent from the right data limit [12].

In the next screen (S11 Fig), the sedimentation velocity analysis workflow with the 2-Dimensional Spectrum Analysis (2DSA) [28, 29] or Custom Grid (CG) analysis [30] methods can be pre-defined for each channel. The user can specify grid options, including predefined custom grids, grid resolution, subgrid size, and two-dimensional grid parameters, including exchange of frictional ratio for partial specific volume as one of the grid dimensions. The analysis flowchart provides default options for fitting of time- and radially-invariant noise as well as boundary conditions, refinements, and Monte Carlo iterations. Meniscus fits can be manually inspected before proceeding, or automatically processed. All default options follow the analysis workflow recommended in [27]. An optional parametrically constrained spectrum analysis [18] with all options available in the non-GMP UltraScan version can be added for each channel (see S12 Fig).

After defining the analysis workflow for each channel, a complete AUC analysis profile has been created, which is saved using a global unique identifier (GUID) in the database. This profile can be indefinitely reused, but any changes will force a name change and a new GUID to be assigned. This prevents any changes to a GMP protocol and assures the same method can be reused for future repeat experiments. In the next screen (S13 Fig), a check is performed to assure that all relevant run parameters have been defined, and can be submitted to the Optima instrument and to the UltraScan LIMS database where it will be stored in read-only mode. Only instrument specific details will be sent to the Optima. In the Optima, the GMP analysis profile will appear as a "Method Scan" entry in the PostgreSQL database. It contains all of the instructions to perform the entire data acquisition process, and cannot be edited from the

Beckman-Coulter Tomcat web interface. Immediately after submission, the user is instructed to load the rotor, select the newly submitted method scan and start the experiment from the instrument. Further, the software will assure that any GMP repeat experiments derived from this profile are launched from the UltraScan software, and are executed under the controlled workflow orchestrated by UltraScan by checking the associated PostgreSQL ID issued by the Optima whenever a protocol has been submitted to the Optima.

**C4. Data acquisition and monitoring.**   After the experimental analysis profile has been submitted, the UltraScan GMP software automatically switches to the "Live Update" panel, monitoring any data acquisition activity occurring on the Optima AUC instrument. As soon as the run is started, temperature and rotor speed monitoring commences (S14 Fig). During data acquisition, experimental data are displayed on the monitor screen and updated in real time. A scan selection panel allows the user to switch between optical systems if both absorbance and interference optics are used, and to switch between cells, channels, and wavelengths (see S15 Fig). The rotor speed and chamber temperature are continuously updated in the GUI to show the current status of the instrument.

**C5. Data import into the UltraScan LIMS database.**   Once all expected data have been retrieved from the Optima AUC instrument, the UltraScan GMP program switches to the "LIMS Import" stage, and uploads the data into the LIMS database (S16 Fig). If intensity data were acquired, the user is able to manually select the air absorbance region above the meniscus, which defines the maximum transmitted light intensity reference for all scans, to be used in the conversion of intensity data to pseudo-absorbance data, using a different reference intensity for each wavelength. Upon conversion to pseudo-absorbance (S17 Fig), the data are uploaded into the UltraScan LIMS database and associated with all relevant metrics from the experiment and the experimental analysis profile definition. It should be stressed that GMP operation benefits from the direct and automated import of experimental data from the instrument into the UltraScan LIMS database in important ways, and has several key advantages compared to the default data import method performed with the Beckman-supplied software: 1. Experimental data stored in the UltraScan LIMS remain in their native double-precision format for all downstream operations, and do not suffer the information loss incurred when data are converted into ASCII formatted data by the Beckman program; 2. The default method for downloading data is tedious, requires special software to uncompress and rename data files before they can be consumed by third-party analysis software; 3. The default ASCII format consumes significantly more disk space and takes more time to read and write; 4. At no time are data manually stored by the user in an intermediate disk storage environment which introduces issues for GMP operation because of the possibility of unauthorized access and alteration of experimental data.

**C6. Data editing.**   After experimental import, an automated editing step is included that defines the meniscus position and the start and end data range for each experimental dataset. The end data range is based on the value entered in the general analysis profile screen (S10 Fig), while the meniscus position is determined with a peak finding algorithm near the column position predicted by the loading volume entered on the same screen. The left experimental data range is automatically identified 0.03 cm to the right of the determined meniscus position to insure boundary conditions will not be set inside the data range during a meniscus fit. Meniscus positions, baseline and plateau values (needed for the van Holde–Weischet [31] and the dC/dt [32] analysis) for each triple are determined automatically during this stage. First, the meniscus position is estimated based on a simple functional analysis of the data scans corresponding to the region in the vicinity of the cell top which, in turn, is estimated based on the parameters defining the centerpiece geometry discussed above (all derived from the LIMS database), and the loading volume for the channel (defined in the analysis_profile). For

example, the initial estimate of the meniscus position, $r_m$, for a sector shaped centerpiece is inferred from the loading volume, $v$ (in μl), the sector angle $\theta$ (in degrees), the bottom position of the cell, $r_b$ (calculated in cm from the rotor stretch profile [12]), and the pathlength $h$ (in cm) of the centerpiece as shown in Eq 1:

$$r_m = \sqrt{r_b^2 - \frac{0.36v}{h\theta\pi}}$$

Eq 1

Each scan in the triple dataset is then examined for the existence of a prominent peak in the immediate vicinity of the initial estimate for the $r_m$ value, which is accepted as the meniscus value for each scan. The actual meniscus position is defined as the average over the meniscus positions from all scans belonging to the same triple. To improve the reliability of the method, only the last ~20% of the scans in the triple are used for this analysis, since later scans in an experiment are expected to have absorbances closer to the baseline near the meniscus and are therefore more easily to recognize as meniscus positions. Next, the baseline of the dataset is determined by taking the average absorbance value of 20 points, centered at the radial position in the last scan of the experiment 0.03 cm to the right of the meniscus position. Similarly, the plateau value for each scan is defined as the absorbance average of 20 points at the right limit of the data range, shifted to the left by a predefined offset value (also 0.03 cm). All computed parameters (meniscus, data range, plateau, and baseline) are recorded in the 'Edit Profile' associated with the corresponding raw dataset (a triple). In the case of a multi-wavelength AUC experiment, these parameters are applied to all triples from all wavelengths associated with the same cell and channel. Optionally, a manual editing window is available (S18 Fig) to fine-tune other parameters of the editing process. At this time, only intensity data are supported, since special cases encountered during interference data editing prevent reliable automation. We hope to arrive at robust algorithms in the future to include interference experiments as well.

**C7. Data analysis.** The data analysis phase is an automated process that sequentially applies the fitting refinement steps defined in the analysis profile above on a remote supercomputer or local cluster. All communications with the cluster use secure sockets layer (SSL) encryption to prevent unauthorized man-in-the-middle access of experimental data or results. The sequence of the data analysis process is defined by the analysis process record and managed by the 'Autoflow' supervisor, a process control system that monitors each step of the GMP data collection and analysis workflow, which is illustrated in S19 Fig. The analysis workflow logic was previously discussed in [27]. Briefly, the default analysis workflow starts with a 2DSA analysis with time-invariant noise removal, followed by a meniscus and bottom fit performed with time- and radially invariant noise removal. Additional refinement is achieved with the iterative 2DSA step, which again includes time- and radially-invariant noise removal, followed either by genetic algorithm analysis [33] for pauci-disperse samples, by 2DSA-Monte Carlo analysis [34], or by PCSA analysis when heterogeneous samples are involved. All models generated by sequential analysis steps are deposited in the UltraScan LIMS database for further meta data analysis or analysis reporting. During analysis, all datasets are submitted to a high-performance computer for parallel analysis, using batch processing [35]. In the Analysis tab, the user can follow the progress of each dataset's analysis and receive real-time job status updates (see S20 Fig, S21 Fig).

**C8. Reporting.** All reporting is performed automatically and is based on the fitted models deposited in the UltraScan LIMS database. Optionally, analysis results can be compared to a validated reference data set. A reference data set has been processed with the same experimental analysis profile, and detailed results are available for comparison. For example, a reference data set may stem from the analysis of a brand name pharmaceutical, or a DNA standard

previously analyzed under GMP conditions. In the report, a list of values obtained from tests performed on the primary data and the fitted models, are compared to the values obtained from the same tests performed on the reference dataset. Alternatively, the values can be reported as-is, if a reference dataset is not available. The criteria tested include properties of the experimental data, such as loading volume, loading concentration, duration of the experiment, rotor speed, number of scans collected, wavelength ranges used for measurement, and the solution column length, as well as the root mean square deviations for the fitted models. In addition, the results from a metadata analysis, where partial concentrations of individual analytes are compared, or percentages of signal observed in subsections of the 2DSA grid, will be reported. Reports for comparisons with reference data sets will provide overlay plots and percent deviations for the tested values. If values exceed the pre-configured tolerance values, the report will be flagged. Thus, the software is able to provide the user with an entirely unbiased analysis report. For custom reports and additional analysis not considered in UltraScan GMP, the same data can also be retrieved (in read-only mode) from the academic UltraScan software and further processed with the full set of functionality available in UltraScan. In the reporting screen (S22 Fig), a report for each dataset can be viewed or downloaded for each dataset, and reports on all parameters discussed above, and their deviations from the set tolerances. A Pass/ Fail grade will be assigned based on whether a data analysis met all previously set tolerance values (see Section **C3**). Instrument specific performance data, such as RMSD values expected from a well-tuned instrument, as well as expected agreement with reference standards will be determined in future experimental work.

**C9. Autoflow Supervisor.** The data acquisition program has a flexible yet simple and intuitive interface to manage experiments submitted to the Optima AUC. It features screens for designing and/or submitting a new experimental analysis profile, to monitor the experimental progress, and to import and post-process acquired data. In addition, it allows the user to re-attach to any stage of this process, and to delete abandoned experiments from the list it manages. The management is performed by the autoflow supervisor, which keeps track of each experiment by linking the details of the designed experimental protocol to the details on the associated experiment that was launched from a particular Optima instrument. The protocol details are represented by an experimental description, a profile name and a unique ID in the Optima and the LIMS database. Experimental profiles include information required for the Optima to execute the profile, such as the duration of the experiment, details on the wavelength range(s), scan frequency, and optical systems used. For laboratories with multiple instruments, the supervisor must also keep track of multiple Optima instruments. This information represents a small subset of the protocol record kept in the LIMS database and is used by the autoflow supervisor whenever protocol details need to be retrieved at different stages of the program execution. To keep track of the stage for each experiment, an autoflow database table has been added to the UltraScan LIMS database, which keeps the state information for real time access, and can be queried by any software instance, at any time for the up-to-date system state of the UltraScan GMP module.

For each experiment, the Optima AUC stores the associated runID (generated upon launching the protocol from the Optima panel), the run status, and the time stamp information. Like the experimental protocol details, the Optima AUC's runID can be used to access a wide range of information about the instrument status or acquired data, as soon as it becomes available. Together, these concise sets of parameters provide a unique signature for any experiment. These run signatures are stored in the autoflow LIMS database table. In addition to the experimental analysis profile's ID and the Optima status information, these table records include a field identifying the state of each experiment, which is used to switch UltraScan GMP to the correct processing stage. For example, when the experiment is submitted to the

Optima instrument, the program switches to the "Live Update" stage, an autoflow table record is generated which includes protocol and run details, and the program status ("LIVE_UP-DATE"). When the Optima run is completed, the program proceeds to the LIMS Import stage, and the autoflow table record for this run is updated with status "IMPORT" (and other data relevant parameters). The autoflow table records are used to both characterize Optima runs, and identify and reattach to the appropriate program's stage, be it monitoring during data acquisition (2. Live Update), importing of the retrieved data into the relational LIMS database (3. LIMS Import), data post-processing during data editing (4. Editing) or data analysis (5. Analysis). A report is generated at the end, concluding the autoflow record. Together, these features allow multiple users to monitor ongoing experiments with random access. To prevent corruption from multiple accesses, a global unique identifier (GUID) is used as a session variable that prevents multiple users from simultaneously importing the same experimental data into the LIMS database, or submitting different experiments to the same instrument. The autoflow process is illustrated in S19 Fig.

## Results and future directions

To this date, the UltraScan GMP software has been implemented on two Optima AUC instruments at the Canadian Center for Hydrodynamics (CCH) at the University of Lethbridge, using the Chinook supercomputer for initial testing and development, and recently for production applications. In addition, the software is in use at over 30 commercial and academic sites throughout the US, the United Kingdom and in Germany. We have performed over 400 experiments at the CCH to validate its stability and suitability for the intended purpose. Our current efforts focus on testing a large range of experimental configurations and scenarios to prepare the software for FDA validation and to make it compliant with ICH Q2 and 21 CFR part 11. We invite participation of industry and developers to further develop AUC validation standards, in particular testing of dsDNA standards on multiple instruments for comparison and establishing reproducibility criteria and fitting statistics which will be needed for future validation of software and hardware, and informing the instrument-specific parameter tolerances relevant for the reporting module. These efforts will guide the development of the fully automated analysis and reporting modules which are still under development at this time. Additional improvements can be achieved in the graphical user interface (GUI) design by developing code to achieve greater separation between the GUI operations and the database operations to achieve a more responsive GUI and to take advantage of modern multi-core architectures. Based on feedback received so far, the UltraScan GMP module, in combination with the Beckman-Coulter Optima AUC, represents a major step forward towards GMP operation of the analytical ultracentrifuge.

## Availability and software dependencies

The UltraScan software is an open-source, multi-platform software suite for hydrodynamic modeling, written in C++ and licensed under the LGPL software license, version 3. It can be downloaded as a pre-compiled binary for Windows, Macintosh and Linux/X11 from ultrascan.aucsolutions.com, or in source code format from GitHub at github.com/ehb54/ultrascan3. The software connects directly to Optima AUC instruments through ethernet networking, and must be connected to a LIMS server and high-performance computing resource (publically available LIMS servers and supercomputers are offered through the NSF-funded XSEDE Science Gateway for hydrodynamic modeling: http://uslims.aucsolutions.com). Testing data are available from the UltraScan workshop resource page at: http://ultrascan.aucsolutions.com/workshop.php. UltraScan relies on the following dependencies, all

of which are available under open source licenses: Qt 5.x (https://doc.qt.io/qt-5/), QWT (https://sourceforge.net/projects/qwt/), MariaDB [25], OpenSSL (https://www.openssl.org/), Qwtplot3D (http://qwtplot3d.sourceforge.net/), Zlib (https://github.com/madler/zlib), and a gcc/g++ compiler version 9.2.0 or newer. The supercomputer and cluster interface require a scheduler and OpenMPI (www.open-mpi.org). UltraScan includes context-specific, online help documentation for each module. Additional documentation is available on the Ultra-Scan3 website, which includes startup instructions and installation videos, tutorials, wikis, links to workshops and troubleshooting information. The UltraScan LIMS software requires a properly configured Linux server. We have tested various Linux distributions, including CentOS 7.x and CentOS 8.x (available from www.centos.org/download), but other distributions can be used and adapted. The Linux servers must be configured with the Apache webserver (apache.org), MariaDB [25], and PHP version 7.x (www.php.net), and must have the Apache Airavata grid middleware layer installed [26]. Additional information on the installation of the Apache Airavata interface can be obtained from GitHub at github.com/apache/airavata and on SciGaP at scigap.org. The GMP analysis module of UltraScan requires access to a supercomputer or cluster, which must be configured with a scheduler, typically provided through PBS (www.pbspro.org) or through torque, available from GitHub at github.com/adaptivecomputing/torque. The LIMS software is also licensed under the LGPL software license, version 3, and can be download from GitHub at github.com/ehb54/ultrascan3_all. The Optima AUC instrument is available from Beckman-Coulter, 5350 Lakeview Parkway South Drive, Indianapolis, IN 46268, USA.

## Supporting information

**S1 Fig. Timestate object viewer in UltraScan.** Actual temperature, rotor speed, scan interval, centrifugal force integral, time, set speed, and run stage are recorded in 1-second intervals and used to verify values recorded in scans
(TIF)

**S2 Fig. Opening screen for UltraScan GMP when an experiment currently in progress is detected.** The user can re-attach to this experiment, or decide to design or load a profile for a different instrument that is currently idle.
(TIF)

**S3 Fig. Laboratory, instrument, rotor and rotor calibration selection screen.** Also, the instrument, the operator performing the experiment is selected, as well as the experiment type. The instrument's connection status is checked by the program automatically.
(TIF)

**S4 Fig. Speed selection screen. One or more speeds can be specified.** Minimum scan intervals are dependent on speed and calculated by the software automatically. Interference and UV/visible absorbance optics have different scan intervals and initial minimum delay times for the first scan that are instrument and speed dependent.
(TIF)

**S5 Fig. Centerpiece and window selection screen.** UltraScan is able to use all rotor positions for data acquisition and does not need to have a counterbalance located in hole 4 or 8, as long as a 3000 rpm radial calibration has already been performed with the counterbalance in place in a previous calibration run. For GMP experiments, the calibration centerpiece can be included instead to validate radial calibration.
(TIF)

**S6 Fig. Each channel is associated with a specific solution, containing one or more analytes and a buffer composition.** Analytes provide a partial specific volume and extinction coefficient to the analysis, buffer composition is needed to estimate density and viscosity for corrections to standard conditions. Additional solution comments can be added, new solutions can be defined.
(TIF)

**S7 Fig. In the Optics screen, each channel will be associated with one or more optical systems.**
(TIF)

**S8 Fig. Dialog for defining wavelength and radial ranges for data collection. Each cell can be selected for a different wavelength or radial range.**
(TIF)

**S9 Fig.** Wavelength selection: The user is prompted with a dialog (A) to choose between manual definition (B) or a selector dialog (C) to define wavelength ranges. The manual selector provides examples for a simple scripting language to define a series of wavelength choices. In the wavelength selector desired wavelength are selected by clicking on them in a GUI.
(TIF)

**S10 Fig. General analysis profile panel. For each loaded sample, a loading concentration ratio, loading volume and the bottom of the analyzed data column are defined.** Concentration ratios and loading volumes can be associated with a tolerance limit for the reporting section. For replicate samples an "Apply to All" button replicates settings of the first sample to the other rows.
(TIF)

**S11 Fig. Two-Dimensional Spectrum Analysis (2DSA) or Custom Grid (CG) workflow definition screen.**
(TIF)

**S12 Fig. Parametrically constrained spectrum analysis (PCSA) selection window.**
(TIF)

**S13 Fig. Submission screen for the experimental analysis profile.** Submitting the run will save the profile in the UltraScan LIMS database in read-only mode, and send parameters required for executing data acquisition to the Optima instrument.
(TIF)

**S14 Fig. Live Update stage shortly after rotor acceleration (green line in lower plot).** Science modules are being calibrated, and data acquisition has not yet started. Temperature and rotor speed are being monitored as soon as the run has been started on the instrument.
(TIF)

**S15 Fig. Active data acquisition updates. Incoming data are displayed in the upper window in real time, and the user can select between multiple triples and optical systems.**
(TIF)

**S16 Fig. LIMS Import screen allowing the user to define a reference point in the region above the meniscus to prompt conversion to pseudo-absorbance data (next Figure) for all data.**
(TIF)

**S17 Fig. Pseudo-absorbance converted intensity data ready to be saved to the LIMS database.**
(TIF)

**S18 Fig. Editing window. Separate interfaces exist for manual editing of absorbance or interference data. During GMP operation, an automatic editing feature is based on values entered previously into the experimental analysis profile.**
(TIF)

**S19 Fig. The Autoflow workflow schema.**
(TIF)

**S20 Fig. Analysis overview of all acquired datasets. Clicking on the triangle in front of the triple's name expands the view to show queuing details for each each analysis (as shown in S21 Fig).**
(TIF)

**S21 Fig. Expanded view of the analysis workflow shown in S20 Fig for a dataset submitted to a high performance computing infrastructure. All scheduled analysis steps are shown, and their current status.**
(TIF)

**S22 Fig. Reporting screen: A pass/fail grade is assigned based on tolerances selected in the experimental protocol, the report can be viewed or downloaded as a PDF file for individual datasets.**
(TIF)

## Author Contributions

**Conceptualization:** Alexey Savelyev, Gary E. Gorbet, Borries Demeler.

**Data curation:** Amy Henrickson.

**Funding acquisition:** Borries Demeler.

**Methodology:** Alexey Savelyev, Gary E. Gorbet, Borries Demeler.

**Project administration:** Alexey Savelyev, Borries Demeler.

**Resources:** Borries Demeler.

**Software:** Alexey Savelyev, Gary E. Gorbet.

**Supervision:** Borries Demeler.

**Validation:** Gary E. Gorbet, Amy Henrickson, Borries Demeler.

**Visualization:** Gary E. Gorbet, Amy Henrickson, Borries Demeler.

**Writing – original draft:** Borries Demeler.

**Writing – review & editing:** Alexey Savelyev, Amy Henrickson, Borries Demeler.

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
