## [Decision Letter · Decision Letter 0]

11 May 2020

Dear Dr. Demeler,

We are pleased to inform you that your manuscript 'Moving Analytical Ultracentrifugation Software to a Good Manufacturing Practices (GMP) Environment' has been provisionally accepted for publication in PLOS Computational Biology.

Best regards,

Dina Schneidman-Duhovny

Software Editor

PLOS Computational Biology

Reviewer's Responses to Questions

**Comments to the Authors:**

Reviewer #1: A very nice introduction of the AUC into the GMP world, as an analytical scientist I think the AUC offers a very powerful resolution that is not exploited enough due to GMP issues, so I am very happy to see that there is progression in this field. I have a few comments and questions but overall I would recommend this publication to be accepted. It would be good to see the inclusion of some fit for purpose parameters for the OPTIMA in the manuscript as i assume the method would need to be validated as per ICH Q2. I guess this will also be 21 CFR part 11 compliant, if approved by the FDA? The manuscript references "acceptable error tolerances" it is un clear who will define these acceptance criteria, the user or ultrascan? please clarify. if Ultra scan will set these criteria will there be statistical tool in bedded into the analysis software when comparing back to reference data, to allow us to conclude if this passes or fails? Or will this be based on whether the user deems it to be comparable? I think there is a benefit in using some sort of statistical approach for the acceptable tolerance. It would be good to include what the final report would look like, this was the only thing that was missing for me. Maybe a follow up publication could include a therapeutic Mab as an example of what the workflow would look like in a GMP setting? (a simple comparability). In section C7 data analysis - it states this is an automated process with an outlined sequence of events my question here is, i guess the experimental parameters that will be fitted (i.e s value range and f/f0 range) will be based on the reference data set that has been defined prior in a non GMP environment?

Reviewer #2: Section B "Validated standards":

The authors propose dsDNA and a radial calibration mask as calibration standards.

Questions:

1) Will you make these items publicly available? Especially the hydrodynamic properties of the DNA may depend on the source (methylation). A common source would be therefore desirable.

2) The diffusion coefficient of DNA quickly becomes very small (and the f/f0 very large) with increasing size and its measurement is often associated with a substantial error. Could you please comment on why you prefer dsDNA over a set of calibration proteins commonly used for SEC?

Sections C7 "Data analysis" and C8 "reporting"

Would it be possible to include an additional supplementary figure for each of these steps in the workflow?

Reviewer #3: Due to substantial recent advances in instrumentation and data analysis software, analytical ultracentrifugation (AUC) has gained increasing interest particularly in biopharmaceutical industry as an orthogonal method for quality assessment and control of their products in addition to established methods, e.g., size exclusion chromatography. However, in order to satisfy regulatory requirements, AUC is obliged to progress toward good manufacturing practice (GMP) compliance.

In their manuscript, the authors have thoroughly analyzed and discussed the current limitations of AUC, and for the first time have proposed a detailed plan for bridging the gap toward GMP conformity by addressing several important compliance aspects of data acquisition and analysis.

On the experimental side, the authors have proposed numerous parameters like temperature, speed and reference standards (e.g., double-stranded DNA) that can easily be cross-validated and employed for validation of proper AUC instrument function.

The authors have also pointed out that a direct connection between data acquisition on Beckman Coulter’s novel Optima AUC and analysis, preferably within the same software program, is crucial for fulfillment of GMP requirements, particularly with regard to ensuring data integrity and consistency. The authors' continuously developed and improved open-source AUC data analysis package, UltraScan, satisfies these prerequisites appropriately. UltraScan now offers a major addition that allows for programming comprehensive, automated workflows in order to minimize operator interactions. These workflows can be adjusted via a well-designed graphical user interface and cover all steps from submission of a complete set of predefined experimental parameters for data acquisition to Beckman Coulter’s Optima AUC to automated data evaluation. All experimental settings (e.g., sample information, instrument and rotor IDs, calibration data, used centerpieces and windows, temperature, speed, wavelength, etc.) and read-outs (e.g., sedimentation profiles and timestates of temperature, speed, ω2t, etc.) are automatically transferred and deposited in the accompanying UltraScan laboratory information management system (LIMS) database. These information will be used for automated data evaluation and reporting according to predefined methods, which encompass checking for acceptance criteria (e.g., loading volume, constancy of temperature and speed, etc.), data editing (i.e., definition of meniscus positions and data fit ranges) and the employment of UltraScan’s ‘traditional’ broad range of advanced data analysis algorithms.

As a non-essential but worthwhile suggestion for improvements, the authors should implement recording of timestates for the vacuum level as an additional diagnostic read-out in UltraScan.

With the submitted manuscript, the authors have made a major and long-awaited contribution to moving AUC, in particular the sophisticated combination of UltraScan and Optima AUC, toward GMP conformity.

**Have all data underlying the figures and results presented in the manuscript been provided?**

Reviewer #1: Yes

Reviewer #2: Yes

Reviewer #3: Yes

PLOS authors have the option to publish the peer review history of their article (what does this mean?). If published, this will include your full peer review and any attached files.

Reviewer #1: Yes: Elizabeth Rodriguez

Reviewer #2: Yes: Alexander Bepperling

Reviewer #3: Yes: Nikola Wenta

---

## [Editor Report · Acceptance letter]

10 Jun 2020

PCOMPBIOL-D-20-00504 

Moving Analytical Ultracentrifugation Software to a Good Manufacturing Practices (GMP) Environment

Dear Dr Demeler,

I am pleased to inform you that your manuscript has been formally accepted for publication in PLOS Computational Biology. Your manuscript is now with our production department and you will be notified of the publication date in due course.

With kind regards,

Laura Mallard
